# Systematic Identification of Long Noncoding RNAs during Three Key Organogenesis Stages in Zebrafish

**DOI:** 10.3390/ijms25063440

**Published:** 2024-03-19

**Authors:** Chune Zhou, Mengting Li, Yaoyi Sun, Yousef Sultan, Xiaoyu Li

**Affiliations:** 1Henan International Joint Laboratory of Aquatic Ecotoxicology and Health Protection, College of Life Sciences, Henan Normal University, Xinxiang 453007, China; 041106@htu.edu.cn (C.Z.);; 2Department of Food Toxicology and Contaminants, National Research Centre, Dokki, Cairo 12622, Egypt

**Keywords:** long noncoding RNAs, three key organogenesis stages, systematic identification, zebrafish, lncRNA *gas5*, muscle development

## Abstract

Thousands of lncRNAs have been found in zebrafish embryogenesis and adult tissues, but their identification and organogenesis-related functions have not yet been elucidated. In this study, high-throughput sequencing was performed at three different organogenesis stages of zebrafish embryos that are important for zebrafish muscle development. The three stages were 10 hpf (hours post fertilization) (T1), 24 hpf (T2), and 36 hpf (T3). LncRNA *gas5*, associated with muscle development, was screened out as the next research target by high-throughput sequencing and qPCR validation. The spatiotemporal expression of lncRNA *gas5* in zebrafish embryonic muscle development was studied through qPCR and in situ hybridization, and functional analysis was conducted using CRISPR/Cas9 (clustered regularly interspaced short palindromic repeats/Cas9, CRISPR/Cas9). The results were as follows: (1) A total of 1486 differentially expressed lncRNAs were identified between T2 and T1, among which 843 lncRNAs were upregulated and 643 were downregulated. The comparison with T3 and T2 resulted in 844 differentially expressed lncRNAs, among which 482 lncRNAs were upregulated and 362 lncRNAs were downregulated. A total of 2137 differentially expressed lncRNAs were found between T3 and T1, among which 1148 lncRNAs were upregulated and 989 lncRNAs were downregulated, including lncRNA *gas5*, which was selected as the target gene. (2) The results of spatiotemporal expression analysis showed that lncRNA *gas5* was expressed in almost all detected embryos of different developmental stages (0, 2, 6, 10, 16, 24, 36, 48, 72, 96 hpf) and detected tissues of adult zebrafish. (3) After lncRNA *gas5* knockout using CRISPR/Cas9 technology, the expression levels of detected genes related to muscle development and adjacent to lncRNA *gas5* were more highly affected in the knockout group compared with the control group, suggesting that lncRNA *gas5* may play a role in embryonic muscle development in zebrafish. (4) The results of the expression of the skeletal myogenesis marker *myod* showed that the expression of *myod* in myotomes was abnormal, suggesting that skeletal myogenesis was affected after lncRNA *gas5* knockout. The results of this study provide an experimental basis for further studies on the role of lncRNA *gas5* in the embryonic skeletal muscle development of zebrafish.

## 1. Introduction

With the increasing development of high-throughput sequencing technology, lncRNAs have been largely discovered in the transcriptomes of humans and several model organisms, including zebrafish. Long non-coding RNA (lncRNA) is a class of non-coding RNAs whose transcript length is not less than 200 bp, and is poorly conserved. Many studies have found that long non-coding RNAs play an important regulatory role in embryonic development, including the differentiation and maintenance of tissues and organs [1,2,3,4,5,6]. They exert their biological functions through epigenetics at the transcriptional and post-transcriptional levels. Recently, Pauli et al. identified 1133 long noncoding RNAs expressed during zebrafish embryogenesis [7]. Another study identified 77 tissue-specific expressed lncRNAs across five investigated adult tissues [8]. In addition, 813 cardiac lincRNA transcripts were identified by performing genome-wide RNA sequencing of zebrafish embryonic hearts, adult hearts, and adult muscle [9]. These studies all suggest that lncRNAs play a certain role in embryonic development and tissue formation. The present study focused on the expression profile of three key organogenesis stages (10, 24, 36 hpf). The hundreds of differentially expressed lncRNAs were screened out across three key stages, during which lncRNA *gas5* caught our attention.

LncRNA *gas5* has direct homologs in humans and mice. LncRNA growth arrest-specific transcription 5 (*gas5*) was first isolated and identified by Schneider et al. in 1988 from the mouse embryonic fibroblast line of growth inhibition, NIH3T3 [10]. According to the literature, lncRNA molecules play an important regulatory role in muscle development and muscle-related diseases [11,12]. LncRNA *gas5* is a key regulatory transcript of mammalian muscle growth and cell apoptosis. Studies have found that lncRNA *gas5* regulates the expression of PTEN by acting on miR-21, thus controlling the proliferation of myocardial fibroblasts and myocardial fibrosis [13]. Several studies have found that lncRNA *gas5* expression is downregulated in some cancer tissues compared to adjacent normal tissue [14,15,16,17,18]. 

In recent years, zebrafish have become the most popular animal model for the study of developmental biology and molecular genetics. As an excellent model species, zebrafish have been widely used because of their numerous advantages, e.g., high fecundity; aptitude for in vitro embryonic development; transparent embryos; convenient real-time imaging; cheap feeding costs; fast growth rate; short breeding cycle; wide variety of established transgenic mutants; and so on [19]. The most important characteristic of zebrafish as an animal model is that 71.4% of zebrafish genes are conserved with human genes [20,21], and the main tissues and organs of zebrafish have similar features to those of humans at the anatomical, physiological, and molecular levels, including the brain [22], heart [23], muscles [24,25], kidney [26], and liver [27].

In this study, zebrafish were selected as the animal model, and the three important organogenesis stages of zebrafish were used for high-throughput sequencing. Thousands of lncRNAs have been discovered, of which lncRNA *gas5* is of interest to us, and its expression and function during the muscle development of zebrafish embryos were studied through qPCR, in situ hybridization, CRISPR/Cas9, and other experimental techniques. This study aims to explore the role of lncRNA *gas5* in the early embryonic muscle development of zebrafish, and to provide an experimental basis for further research on its molecular mechanisms.

## 2. Results

### 2.1. Quality Assement of High-Throughput Sequencing Data

The higher the quality of sequencing, the higher the quality of subsequent data analysis, and the more accurate and detailed analysis results. The useless data were screened out, and the distribution of read quantity in data quality control is shown in Appendix A. The quality of the sequencing data is summarized in Appendix A.

### 2.2. Identification of Differentially Expressed Genes and lncRNAs at Three Organogenesis Stages

Differentially expressed genes (DEGs) and lncRNAs were identified, and the absolute value of log2FC >1 and *p* value < 0.05 were taken as the criteria (Appendix A). The results showed that a total of 1486 differentially expressed lncRNAs and 5569 DEGs were found between T2 and T1, among which 843 lncRNAs and 4051 DEGs were upregulated and 643 lncRNAs and 1518 DEGs were downregulated (Figure 1). Between T3 and T2, a total of 844 differentially expressed lncRNAs and 2827 DEGs were identified, among which 482 lncRNAs and 2003 DEGs were upregulated and 362 lncRNAs and 824 DEGs were downregulated (Figure 1). A total of 2137 differentially expressed lncRNAs and 8028 DEGs were discovered after comparing T3 and T1, among which 1148 lncRNAs and 5180 DEGs were upregulated and 989 lncRNAs and 2848 DEGs were downregulated (Figure 1).

### 2.3. GO Term and KEGG Pathway Enrichment Analysis of DEGs

To identify the function of DEGs, the DEGs were subjected to GO term and KEGG pathway enrichment analyses [28]. GO term analysis showed that the biological functions of DEGs were mainly concentrated in tissue development, somite development, somitogenesis, embryo development, muscle structure development, etc. (Figure 2, Appendix A).

KEGG pathway enrichment analysis showed that DEGs were mainly involved in the Wnt signaling pathway, ECM–receptor interaction, vascular smooth muscle contraction, MAPK signaling pathway, tight junction, etc. (Appendix A).

### 2.4. qPCR Validation

To validate the reliability of the sequencing data, the expression profiles of 11 differentially expressed lncRNAs and 8 DEGs were detected using qPCR. The qPCR results showed that the expression patterns of DEGs and lncRNAs at three different stages were similar with the RNA sequencing data, indicating that the sequencing data were very reliable and could be used for subsequent studies (Appendix A). Among the discovered differentially expressed long non-coding RNAs, *gas5* has been studied in some other species, including mice and humans, but has not been reported in zebrafish; therefore, the function of the lncRNA *gas5* in zebrafish caught our interest.

### 2.5. The Spatiotemporal Expression Pattern of lncRNA gas5 at Different Developmental Stages of Zebrafish Embryos

The temporal and spatial expression of lncRNA *gas5* was detected by qPCR and whole embryo in situ hybridization at different developmental stages in zebrafish. qPCR results showed that lncRNA *gas5* was universally expressed at different developmental stages of zebrafish embryos (0, 2, 6, 10, 16, 24, 36, 48, 72, 96 hpf), and the expression levels of 6~36 hpf were relatively high compared with other stages (Figure 3b). In situ hybridization results showed that lncRNA *gas5* showed a generalized expression pattern at 0~16 hpf (Figure 3a). However, lncRNA *gas5* was mainly expressed in the eyes, brain, and spinal cord at 24~48 hpf (Figure 3a). At 72 hpf, lncRNA *gas5* was mainly expressed in the head, including the telencephalon, diencephalon, midbrain, posterior brain, cerebellum, and eyes (Figure 3a).

### 2.6. Expression of lncRNA gas5 in Different Tissues of Adult Zebrafish

The expression profiles of lncRNA *gas5* in eight different tissues of adult zebrafish (eye, tail, heart, muscle, liver, kidney, brain, and spleen) were detected by qPCR. The results showed that lncRNA *gas5* was expressed in all the different tissues of adult zebrafish, and the expression level from high to low was as follows: brain > eye > kidney > spleen > muscle > tail > liver > heart (Figure 3c).

### 2.7. The Validation of lncRNA gas5 Knockout

At twenty-four hours after injection, eight eggs were selected for genomic sequencing to test the knockout effect. Sequencing results showed that, compared with zebrafish embryos without injection (NI) and zebrafish embryos only injected with Cas9 protein, the genomic DNA sequencing map of zebrafish embryos co-injected with sgRNA and Cas9 showed homogeneous cross-peaks after the target sites, while no cross-peaks appeared in the control groups, indicating that lncRNA *gas5* knockout was initially successful (Appendix A).

To further verify the knockout results, zebrafish embryos microinjected 24 h later were collected, and the expression level of lncRNA *gas5* was detected by qPCR and in situ hybridization. qPCR detection results showed that the expression level of lncRNA *gas5* in the knockout group was significantly decreased compared with that in the control group (Figure 4a). In situ hybridization results also showed that the lncRNA *gas5* expression levels in brains, eyes, and spinal cords were decreased in the knockout group compared with the control groups (Figure 4b).

### 2.8. Effect of lncRNA gas5 Knockout on Embryonic Development of Zebrafish

After depletion of lncRNA *gas5* using CRISPR/Cas9 technology, the mortality and malformation rates of zebrafish embryos were analyzed. The results showed that the normal survival rate of embryos in the NI group was 93%, the mortality rate was 7%, and there was no developmental deformity. In the group microinjected only with Cas9, 64% of the embryos did not show any phenotypic abnormalities, 34% of the embryos died, and the remaining 2% of the embryos showed developmental abnormalities to some extent. In the knockout group, 55% of embryos died, 3% were deformed, and only 42% survived normally. In conclusion, the death rate and abnormality rate of embryos in the knockout group were higher than those in control groups (Figure 4h), which indicates that lncRNA *gas5* is involved in embryonic development.

Compared with the control group (Figure 4c), delayed embryonic development was observed in the knockout group (Figure 4d), and developmental malformations were also observed to some extent, including enlargement of the yolk extension (Figure 4e), small or absent eyes (Figure 4f), nonseparation of two otoliths, etc. (Figure 4g).

### 2.9. Effect of lncRNA gas5 Knockout on the Expression of Genes Related to Embryonic Muscle Development in Zebrafish

After lncRNA *gas5* was knocked out with CRISPR/Cas9 technology, qPCR was used to test the expression levels of 11 genes (*mef2ca*, *nkx2.5*, *myhc4*, *srfa*, *igf2a*, *myod1*, *myog*, *myf5*, *pax3a*, *pax7a*, *myf6*) related to embryonic muscle development in zebrafish. The results showed that the expression levels of *mef2ca*, *myhc4*, *myod1*, *nkx2.5*, *myf6*, *pax3a*, and *pax7a* in the knockout group were significantly decreased compared with those in the control group. The expression levels of *myog*, *myf5*, and *srfa* were significantly increased compared with that in the control group. The expression level of *igf2a* showed no significant difference compared with the control group (Figure 5a). In summary, lncRNA *gas5* depletion affected the expression levels of some muscle-related genes, which indicates that it may play a certain role in muscle development. However, the specific molecular mechanisms need further study.

### 2.10. Effect of lncRNA gas5 Knockout on the Expression of Its Candidate Target Genes

According to relevant studies, lncRNAs can affect the expression of their neighboring genes through signaling, guiding, isolating, or scaffold molecules [29,30,31,32,33,34]. The prediction of cis target genes of lncRNA *gas5* showed that five genes (*tor3a*, *osbpl9*, *calr*, *si:ch211-198n5+11*, and *rpe65b*), located 100 kb upstream and downstream of lncRNA *gas5*, were identified as candidate target genes. QPCR was used to detect the expression of its five candidate target genes (*tor3a*, *osbpl9*, *calr*, *si:ch211-198n5+11*, *rpe65b*) in zebrafish embryos 24 hpf, after lncRNA *gas5* was knocked out using CRISPR/Cas9 technology (Figure 5b, Appendix A). Interestingly, the results showed that, except *osbp19*, the expression levels of the other four genes decreased significantly compared with that in the control group. This indicates that with the loss of lncRNA *gas5* function, the expression levels of its neighboring genes were affected, but the specific mechanisms involved need to be studied further.

### 2.11. The Skeletal Muscle Development of Zebrafish Was Affected after lncRNA gas5 Was Knocked Out

In order to further study the role of lncRNA *gas5* in zebrafish skeletal muscle development, the expression of the skeletal myogenesis marker *myod* was analyzed using whole-mount in situ hybridization in zebrafish embryos at 10, 24, and 36 hpf. The results demonstrated that, compared with the control embryos (embryos with nothing injected (Figure 6a) and those with only Cas9 injected (Figure 6b)), lncRNA *gas5* knockout led to the abnormal expression of *myod* in the myotomes of KO embryos at 10, 24, and 36 hpf (Figure 6c), which indicates that zebrafish skeletal myogenesis was affected after lncRNA knockout and further proves the role of lncRNA *gas5* in the muscle development of zebrafish.

## 3. Discussion

Several previous transcriptome studies have identified more than 3000 lncRNAs in both embryogenesis stages and adult tissues of zebrafish [7,8,9]. However, the functions of most lncRNAs remain largely unknown. Recently, a number of studies have demonstrated that lncRNAs are related to cardiovascular development [35,36], heart development [37,38], and neurodevelopment [39,40]. Genes preferentially expressed in one of the stages could be of high relevance to stage-specific developmental and molecular events [41]. The morphology of zebrafish changes greatly across 10, 24, and 36 hpf—the rudiments of the primary organs become visible, the tail bud becomes more prominent at segmentation stages (10~24 hpf) [42], and embryos at these three stages undergo key organogenesis stages. Genome-wide sequencing was performed to obtain genes associated with specific organogenesis processes, and our interest focused on lncRNA *gas5*.

LncRNA *gas5* has homologs in humans and mice [43], and some studies have demonstrated that it plays an important regulatory role in muscle development and proliferation in myocardial and muscle-related diseases [11,12,13]. In addition, several studies on lncRNA *gas5* have focused on tumor biology, and most of the research results show that the relative expression of lncRNA *gas5* in tumor tissues is decreased, indicating that lncRNA *gas5* has certain anticancer effects. LncRNA *gas5* was also found to inhibit the activation of cardiac fibroblasts. However, relative research in zebrafish is very limited. Only one report demonstrates that the decrease of lncRNA *gas5* expression results in significantly increased expression of its neighboring gene, *osbpl9* (encoding lipid-binding protein) [44], suggesting that lncRNA *gas5* may play a cis-regulatory role. There is no relevant study on the role of lncRNA *gas5* in the muscle development of zebrafish.

Skeletal muscle is an important muscle type that plays an important role in the life activities of organisms. All skeletal muscle is derived from the corresponding region of the head mesoderm or somatic mesoderm [45,46]. In addition, cardiac muscle and smooth muscle are also important muscle types [7]. Skeletal muscle is an indispensable tissue in animals [47,48]. The developmental process of zebrafish skeletal muscle mainly involves the differentiation of myogenic ganglion cells into myoblasts and the fusion of myoblasts into muscle tubes, which further grow and differentiate into multinucleated muscle fibers under the interaction of a series of signaling pathway molecules and myogenic regulatory factors [49,50]. Myogenic regulatory factors (MRFs) play an important role in skeletal muscle differentiation [51] and are essential in promoting muscle cell differentiation. The MRF family includes four transcription factors: myogenic differentiation antigen (*myod*), *myf5*, myogenin (*myog*), and (*mrf4*). In skeletal muscle development, transcription factors paired box 3 (*pax3*), transcription factors paired box 7 (*pax7*), and MRFs constitute an important regulatory network [52,53]. (MEF2) is a family of MADS frame transcription factors, including MEF2A, MEF2B, MEF2C, and MEF2D. MEF2 and MRF gene families can interact directly to synergistically regulate muscle growth and differentiation through transcriptional activation of the *myog* gene. *Mef2* expression after *myog* can promote myogenic differentiation [54,55]. MEF2 itself does not have the ability to regulate muscle differentiation, but plays an important role in assisting MRFs. In the present study, genes involved in the embryonic muscle development of zebrafish were analyzed using qPCR after lncRNA *gas5* was knocked out, and it was found that their expression levels were changed. Among these detected genes, including some genes associated with muscle development (*mef2ca* [56], *Nkx2.5* [57], *myhc, srfa*), myogenic regulators (*myog*, *myod1*, *myf5*, *myf6*) [58,59,60,61] and transcription factors (*pax3a*, *pax7a*) are all required for skeletal muscle growth and differentiation. Changes in the expression levels of these genes after lncRNA *gas5* was knocked out indicate that the muscle development and differentiation were affected. This results suggest that lncRNA *gas5* may be involved in embryonic muscle development in zebrafish.

The various functions of lncRNAs can be summarized into four molecular mechanism models, which are signal; guide; decoy; and scaffold, in which lncRNAs act as a guide to regulate the expression of neighboring genes through cis-action and affect the expression levels of remote genes through trans-action [34]. In the present study, qPCR was used to detect the expression of five candidate target genes after lncRNA *gas5* was knocked out. The results showed that the expression of the neighboring gene *osbp19* increased significantly compared with that in the control group, which indicates that lncRNA *gas5* may have a cis-regulatory effect on the embryonic development of zebrafish. These results are consistent with previous results [44]. However, Mehdi et al. observed that the transcription of neighboring genes was significantly altered in some lncRNAs-knockout mutants generated by CRISPR-Cas9. But it did not affect embryogenesis, viability, or fertility, which indicates that lncRNAs have no overt function [44]. Therefore, the specific function of lncRNA *gas5* needs to be studied further in the future.

## 4. Materials and Methods

### 4.1. Zebrafish Husbandry and Embryo Collection

The AB wild-type zebrafish (Danio rerio) used in this study were purchased from the Institute of Hydrobiology at the Chinese Academy of Science (Wuhan, China). The experimental fish were bred and maintained in the standard zebrafish breeding system of our laboratory, with a photoperiod of 14 h light/10 h dark in a closed loop aquaculture system according to standard procedures [62]. They were fed twice a day with live brine shrimp. All animal processing was approved by the Institutional Animal Care and Use Committee of Henan Normal University.

On the night before the zebrafish embryos were collected, one female and one male were placed overnight on each side of the baffle in incubation tanks. The baffles were removed in the morning on the second day. Embryos were collected after spawning ended, and were cultured in E3 medium (5 mmol/L NaCl, 0.17 mmol/L KCl, 0.33 mmol/L CaCl_2_, and 0.33 mmol/L MgSO_4_, pH 7.2) at 28 °C. Embryos were harvested according to the experimental requirements, and the different stages of embryonic development of zebrafish were described according to a previous report [42]. A total of 100 embryos from each stage (10 hpf, 24 hpf, and 36 hpf) were selected and randomly divided into two groups for sequencing by the DNBSEQ-T7 sequencer (MGI Tech Co., Ltd., Wuhan, China). The chorionic membrane was manually removed with tweezers and frozen with liquid nitrogen before sequencing.

### 4.2. cDNA Library Contruction, Transcriptome Sequencing, and Bioinformatic Analysis

RNA was isolated from the 3 different key organogenesis stages (10, 24, and 36 hpf) using TRIzol reagent (Invitrogen, Carlsbad, USA, cat. NO 15596026). RNA quality was determined by examining A260/A280 with the NanodropTM OneCspectrophotometer (Thermo Fisher Scientific Inc, Massachusetts, USA.). RNA Integrity was confirmed by 1.5% agarose gel electrophoresis. Qualified RNA was finally quantified by Qubit3.0 with the QubitTM RNA Broad Range Assay kit (Life Technologies, Carlsbad, USA, Q10210). Next, 2 μg total RNA was used for stranded RNA sequencing library preparation using the Ribo-off rRNA Depletion Kit (Catalog NO. MRZG12324, Illumina, Santiago, USA) and the KC-DigitalTM Stranded mRNA Library Prep Kit for Illumina^®^ (Catalog NO. DR08502, Wuhan Seqhealth Co., Ltd., Wuhan, China), following the manufacturer’s instructions.

High-throughput sequencing was performed using the DNBSEQ-T7 sequencer (MGI Tech Co., Ltd., Wuhan, China) with the PE150 model. After the library was constructed, strict quality inspection was carried out on the constructed library. The original sequencing data contained adapter sequences and low-quality read segments, which had a great impact on the subsequent analysis. To ensure the accuracy of the subsequent analysis, the original sequencing data were filtered first. The specific quality control protocol is as follows: remove the adapter sequence in reads and filter out reads shorter than 15 bp. Low-quality reads are filtered and discarded when the proportion of bases below Q20 is greater than 0.08. STAR software (version 2.5.3a) was used to map the reads to the genome. Reads mapped to the exon regions of each gene were counted by feature counts (Subread-1.5.1; Bioconductor), and then RPKM was calculated. Genes differentially expressed between groups were identified using the edgeR package (version 3.12.1). The absolute value of log2FC > 1 and *p* value < 0.05 were used as a standard for differentially expressed genes. Genes located 100 kb upstream and downstream of the lncRNA locus were selected as candidate cis-target genes. Gene ontology (GO) analysis and Kyoto Encyclopedia of Genes and Genomes (KEGG) enrichment analysis for target genes were both implemented by KOBAS software (version: 2.1.1) with a *p*-value < 0.05 as a cutoff.

### 4.3. qPCR Detection

To validate the reliability of sequencing data, the expression profiles of 8 DEGs (*smtnl1*, *tor3a*, *osbpl9*, *calr*, *rpe65b*, *si:ch211-198n5+11*, *tnni1al*, *pvalb4*) and 11 differentially expressed lncRNAs (LOC108179067, LOC108179796, LOC108192140, LOC101885256, LOC100536039, LOC101884451, LOC101884456, LOC103910859, LOC103909205, ch73-21g5.7, lncRNA *gas5*) were detected using qPCR. In addition, to detect the effects of lncRNA *gas5* knockout on the expression of genes related to embryonic muscle development and genes adjacent to lncRNA *gas5*, PCR was also performed. All primer sequences are shown in Appendix A. The reference gene used in this study is eukaryotic translation elongation factor 1 alpha 1 (*eef1a1*), which is used to normalize the initial concentration difference of samples [63]. The reaction system was as follows: 1 μL cDNA, 5 μL MonAmp^TM^ SYBR Green qPCR Mix, 0.4 μL each primer (20 μM), and 3.2 μL nuclease-free water. The PCR conditions were as follows: 10 min at 95 °C, followed by 45 cycles at 95 °C for 15 s, 60 °C for 60 s, and a cooling stage at 4 °C. The 2^−ΔΔCt^ method was used to analyze expression levels.

### 4.4. Zebrafish LncRNA gas5

Through the online Ensembl database (http://asia.ensembl.org/index.html) (accessed on 23 October 2023), NCBI (https://www.ncbi.nlm.nih.gov/) (accessed on 23 October 2022), ZFIN (http://zfin.org/)(accessed on 23 October 2022), and CPAT (http://lilab.research.bcm.edu/cpat/)(accessed on 23 October 2022), the bioinformatics analysis of lncRNA *gas5* found that lncRNA *gas5* is located on chromosome 8 of zebrafish with 14 exons and the transcript length is 799 bp, belonging to the long intergenic noncoding RNA (Appendix A). Analysis results of CPAT (version 1.2.4), an online coding capability software, showed that lncRNA *gas5* does not have the ability to encode proteins (Appendix A).

### 4.5. The Temporal and Spatial Expression Pattern of lncRNA gas5

To further investigate the function of lncRNA *gas5*, the spatiotemporal expression pattern of lncRNA *gas5* was detected by qPCR and in situ hybridization. The prepared RNA was reverse-transcribed into cDNA using the HiFiScript cDNA Synthesis Kit (Cwbiotich, Beijing, China), and the prepared cDNA was diluted 5 times and used as a template for qPCR detection using MonAmp^TM^ SYBR Green qPCR Mix (Monad, Zhuhai China). The reaction system and PCR conditions were the same as above. The primers of PCR and the probe are presented in Appendix A.

The spatial expression of lncRNA *gas5* was detected by in situ hybridization. The PCR fragments were cloned into a pGEM-T easy vector and sequenced to verify whether the fragment was ligated to the vector. The antisense and sense probes were transcribed by SP6 or T7 to generate the DIG-labeled probe using the linearized vector, including PCR target fragments as a template. Whole-mount in situ hybridization was performed as previously described [64].

### 4.6. LncRNA gas5 Knockout by CRISPR/Cas9

CRISPR‒Cas9 sgRNAs (single-guide RNAs) were designed by using the online website (https://www.CRISPRscan.org/?page=sequence) (accessed on 2 March 2023). LncRNA *gas5* has a total of 14 exons, and the target site is located in the sixth exon with a sequence of 20 bp base: GGGATCAAGCTGTATGGAGA (Appendix A). SgRNA was transcribed using the sgRNA In Vitro One-Step Transcription Kit (Novoprotein Scientific Inc., Shanghai, China). The detection primer sequence was as follows: F (5′-3′): CTGAAGCACCATAAACCAAT, R (5′-3′): TAGGCAATAAGCCAGAACAA, and the amplified product length was 438 bp. SgRNA (100 ng/μL) for lncRNA *gas5* and Cas9 nuclease (300 ng/μL) (Novoprotein Scientific Inc., Shanghai, China) were co-injected into the yolks of one- to two-cell embryos, and embryos microinjected with only 300 ng/μL Cas9, as well as those with nothing injected, were regarded as controls. The correct size of the PCR fragment amplified by the detection primer was selected to be sequenced to detect the knockout effect. If the sequencing results show that there are low uniform cross-peaks after the target site, but no cross-peaks before the target site, the knockout is basically successful.

### 4.7. Prediction of lncRNA gas5 Target Genes of cis Action

The prediction of cis target genes is based on the correlation between the function of lncRNA and its adjacent protein-coding genes, and lncRNAs located upstream and downstream of coding proteins may interact with promoters or other cis-acting elements of co-expressed genes, thereby regulating gene expression at the transcriptional or post-transcriptional level. Therefore, we identified the genes within 100 kb upstream and downstream of the lncRNA locus, and these were selected as candidate target genes.

### 4.8. Statistical Analysis

The qPCR experiment was carried out in triplicate. Data are described as means ± standard error of mean (SEM). Shapiro–Wilk tests were used to determine the normal distribution and homoscedasticity. Comparisons of the gene expression levels between control groups and knockout groups were performed by unpaired *t*-tests using GraphPad Prism 8.0. Comparisons of the gene expression levels of different developmental stages and different tissues were performed using one-way ANOVA. Probability (*p*) < 0.05 was deemed as statistically significant.

## 5. Conclusions

In this study, we conducted transcriptome sequencing on three different organogenesis stages of zebrafish embryos, screened thousands of differentially expressed genes and long non-coding RNAs, and further studied the function of lncRNA *gas5*. Firstly, qPCR and whole embryo in situ hybridization were used to study lncRNA *gas5* expression profiles in different developmental stages of zebrafish embryos and in different tissues of adult zebrafish, and the results showed that lncRNA gas5 was expressed in almost every developmental stage and in every tissue. Then, CRISPR-Cas9 technology was used to study its function; the results found that zebrafish embryonic development was affected, and the expression of muscle-related genes and candidate target genes was also affected after lncRNA *gas5* was knocked out. The whole-mount in situ hybridization analyses of *myod* expression showed that the *myod* expression in myotomes was affected after lncRNA *gas5* knockout. All these results indicate that lncRNA *gas5* may play a certain role in the development of zebrafish embryonic skeletal muscle, but its molecular mechanisms need to be confirmed in the future.

## Figures and Tables

**Figure 1 ijms-25-03440-f001:**
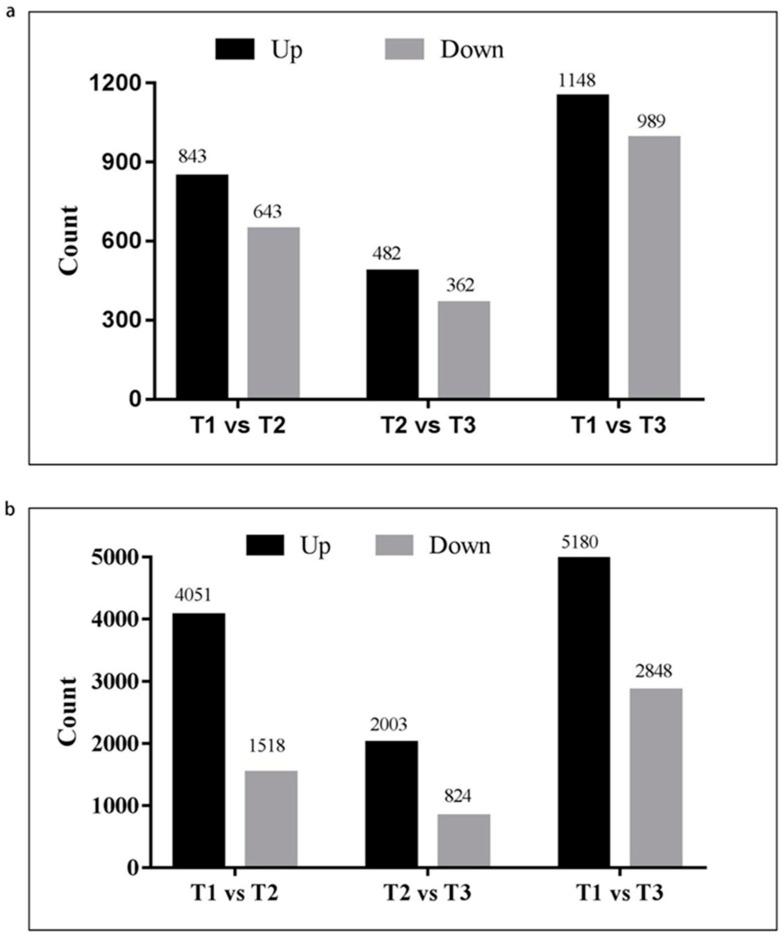
Analysis of differentially expressed mRNAs and lncRNAs in three stages of zebrafish embryonic development. (**a**) Differentially expressed lncRNAs. (**b**) Differentially expressed mRNAs.

**Figure 2 ijms-25-03440-f002:**
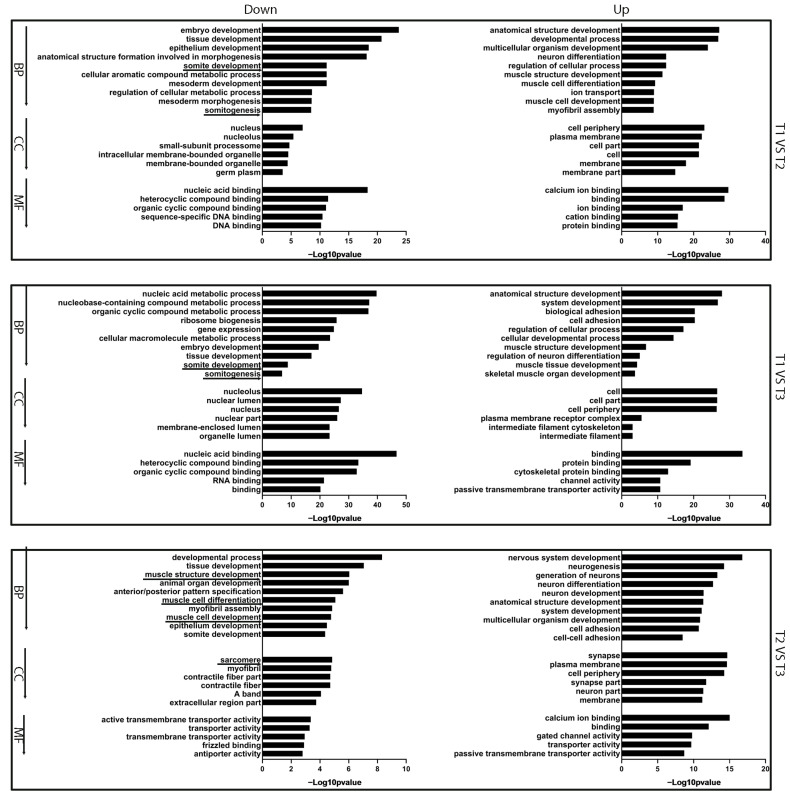
GO enrichment analysis of DEGs at different stages of zebrafish embryonic development. BP: biological process. CC: cellular component. MF: molecular function. Down means downregulated genes, Up means upregulated genes. Underline indicates the terms related with muscle development.

**Figure 3 ijms-25-03440-f003:**
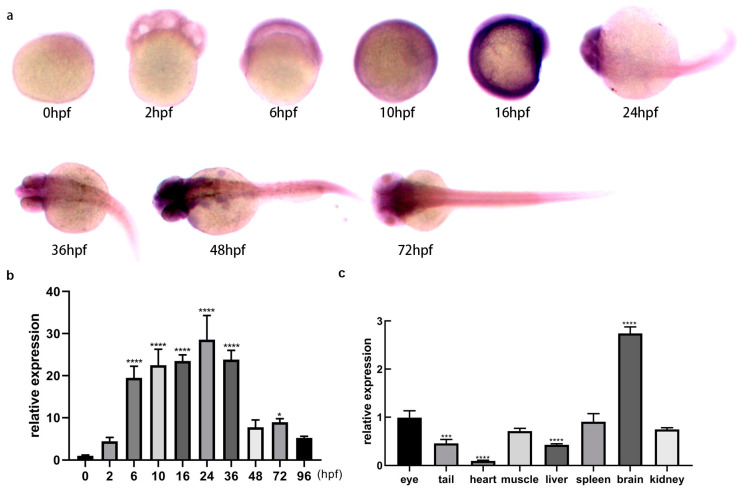
Spatiotemporal expression of lncRNA *gas5* in zebrafish. (**a**) Spatial expression at different developmental stages of zebrafish embryos using in situ hybridization. Microscope’s magnification: 45 ×. (**b**) Expression at different developmental stages, analyzed using qPCR. The expression level of 0 hpf was set as 1, and the relative expression levels of other stages were compared with that of 0 hpf. (**c**) Expression profiles in various tissues of adult zebrafish, detected using qPCR. The expression level of the eyes was set as 1, and the relative expression levels of other tissues were compared with that of the eyes. Data of qPCR derived from three independent experiments are shown as mean ± SEM. * indicates *p* < 0.05, *** indicates *p* < 0.001, **** indicates *p* < 0.0001.

**Figure 4 ijms-25-03440-f004:**
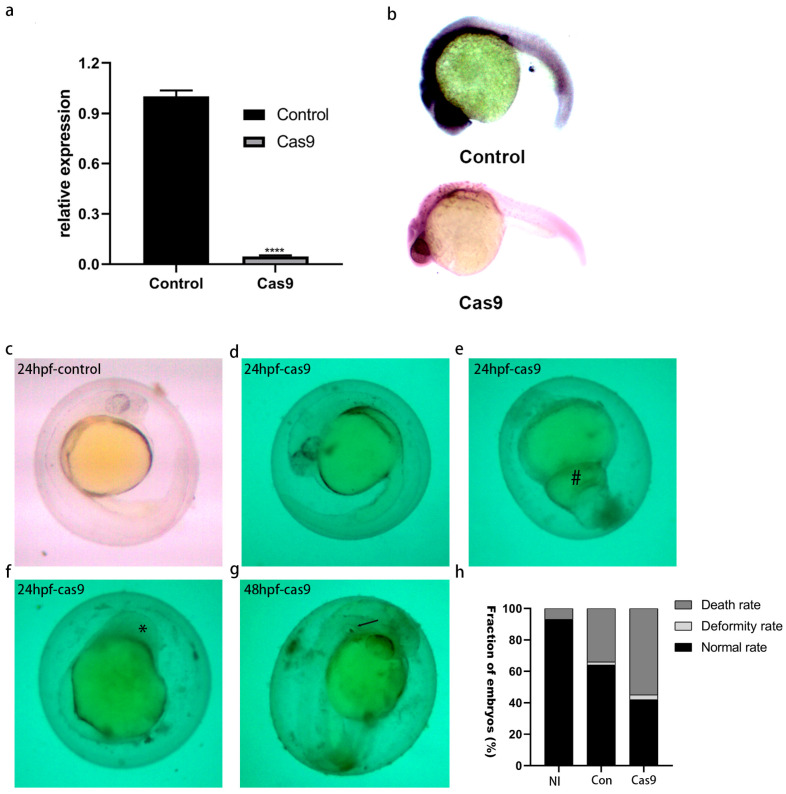
The effects of lncRNA *gas5* knockout. (**a**) The expression level of lncRNA *gas5* knockout embryos, detected by qPCR. Data derived from three independent experiments are shown as mean ± SEM. **** indicates *p* < 0.0001. (**b**) The expression level of lncRNA *gas5* knockout embryos, detected by in situ hybridization. NI represents no injection, control represents only injection of Cas9 protein, Cas9 represents co-injection of sgRNA and Cas9 protein. Microscope’s magnification: 45 × (**c**–**h**) The effects of knockout on embryonic development. Microscope’s magnification: 45 ×. #: enlargement of yolk extension. *: small or absent eyes, arrows indicate nonseparation of two otoliths.

**Figure 5 ijms-25-03440-f005:**
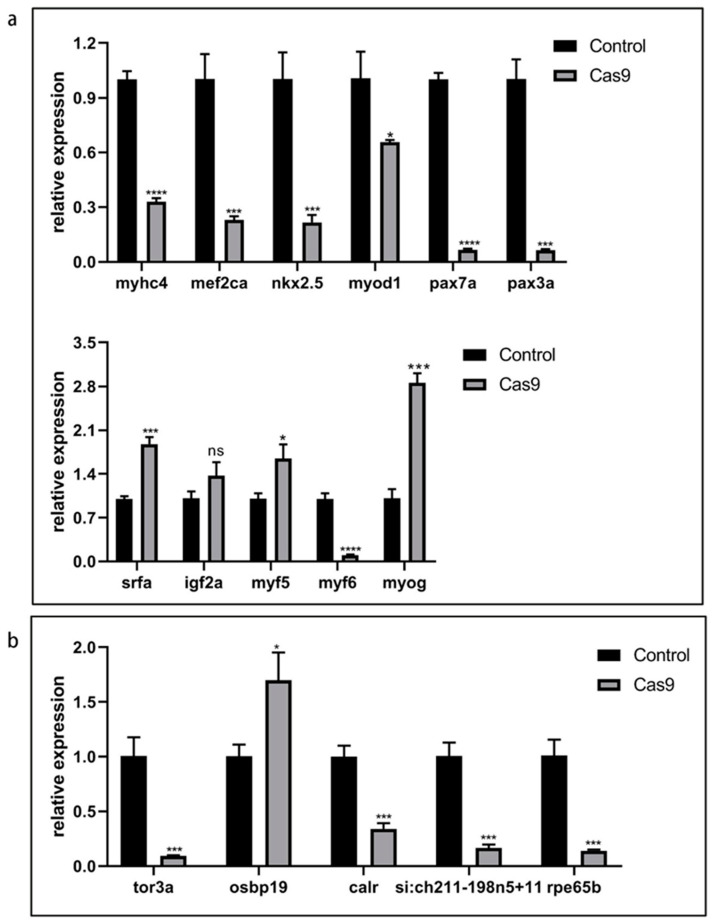
Effect of lncRNA *gas5* knockout on expression levels of related genes. (**a**) Expression of genes related to muscle development in zebrafish embryos 24 hpf. (**b**) Expression of genes adjacent to lncRNA *gas5* in zebrafish embryos 24 hpf. Control represents only injection of Cas9 protein. Data from qPCR derived from three independent experiments are shown as mean ±SEM. * indicates *p* < 0.05, *** indicates *p* < 0.001, **** indicates *p* < 0.0001.

**Figure 6 ijms-25-03440-f006:**
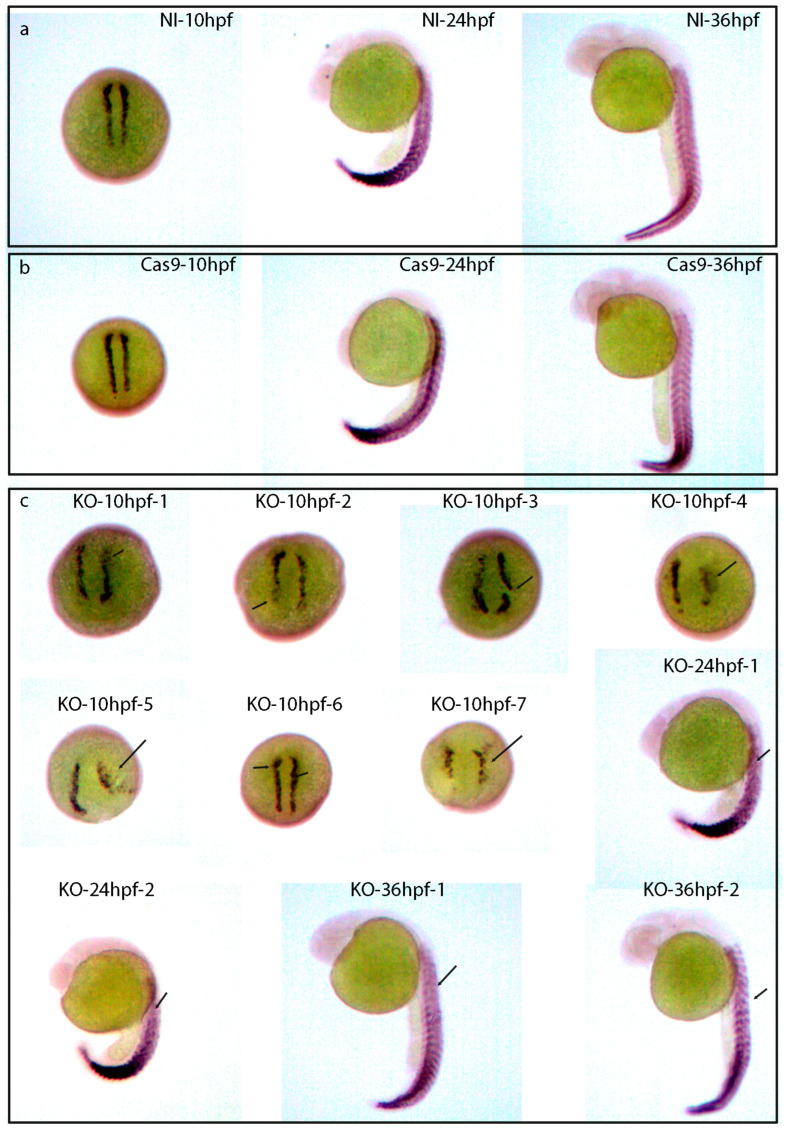
Whole-mount in situ hybridization of skeletal myogenesis marker *myod* in control and knockout zebrafish embryos at 10, 24, and 36 hpf. (**a**) Whole-mount in situ hybridization of skeletal myogenesis marker *myod* in non-injected (NI) embryos at 10, 24, and 36 hpf. (**b**) Whole-mount in situ hybridization of skeletal myogenesis marker *myod* in only Cas9-injected embryos at 10, 24, and 36 hpf. (**c**) Whole-mount in situ hybridization of skeletal myogenesis marker *myod* in knockout embryos (KO) at 10, 24, and 36 hpf. Arrows indicate the abnormal expression of *myod* in myotomes. NI indicates no injection. KO indicates knockout. Microscope’s magnification: 45 ×

## Data Availability

All materials and analyzed data are included in this published article and the Appendix A. RNA-seq data sets used in our study are available in the National Center for Biotechnology Information Gene Expression Omnibus under the accession number GSE235668.

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
