# Peer review of "Systematic Identification of Long Noncoding RNAs during Three Key Organogenesis Stages in Zebrafish"

_ijms, 2024, doi:10.3390/ijms25063440_

Round 1

Reviewer 1 Report

Comments and Suggestions for Authors

In this study, Zhou and coworkers identified long noncoding RNA (lncRNA) expressed at three crucial developmental stages for zebrafish muscle development. The initial phase of the study revealed the presence of differentially expressed mRNAs and differentially expressed long noncoding RNA during these developmental stages. Subsequently, the focus shifted to gas5, a specific lncRNA previously recognized in mammals for its significance in muscle development. The research showcased the role of lncRNA gas5 in embryonic muscle development through diverse technical approaches.

This paper holds significant implications as it strives to enhance our understanding of lncRNA functions across various developmental processes.

In my view, a few limitations are associated with the exclusive reliance on the analysis of F0 animals and the absence of detailed information on the technical approach to CRISPR-CAS 9 technology and other major points.

Suggestions and concerns are outlined below:

-       The authors analyze both transcriptomic data and lncRNA data at three developmental stages, but it is not clear the rationale and the link between them. Explain better.  Moreover, I think it could be interesting to show in which developmental stages of the three analyzed there are major biological processes or KEGG pathways linked to muscle organogenesis, development, and differentiation. Finally, I recommend including a Venn diagram and a heat map in the main paper illustrating the most differentially expressed mRNAs and lncRNAs at the three developmental stages.

-       In the material and methods (lines 357-358) the authors declare to use the absolute value of |logFC| >2 and p value<0.05 for DEG and lncRNA, but results indicate |logFC| >1 (see lines 93 and 94).

- Functional analysis of lncRNA gas5 using CRISPR/Cas9 technology lacks detailed explanation, and results are at times unclear:

a. I suggest providing a schematic drawing displaying the gene architecture of lncRNA gas5 with the target site of the SgRNA.

b. Confirm the sequence for lncgas5 knockout in F0 adult zebrafish and present the alignment of sequences between control and cas9 knockout. The Supplementary Figure S6 is not sufficient to demonstrate the knockout effect and therefore confirm that the observed phenotypic effects are real.

c. Clarify toxic effects observed in Figure 4h, especially the increased mortality rate in embryos with only Cas9 protein injection. Report the number of embryos used in this analysis.

d. Include RT-PCR results using control with only injection of Cas9 protein, not just the control (Figure 4a).

e. Enhance resolution in Figure 4 embryo images to highlight defects, especially regarding otoliths.

f. In Figure 4b I should see a total absence of gas5 expression in the ISH as in the RT-PCR experiments, not a decrease as reported in line 188. Explain better the results obtained.

g. Due to unusual mortality in Cas9 protein-only sample, the authors must present RT-PCR results for genes related to embryonic muscle development (include the sample in Figure 5).

h. Explain in detail the differences between phenotypes observed in Figure 6 (e.g., KO-10 hpf-6 and KO-10 hpf-7, KO-36 hpf-1 and KO-36 hpf-2). Clarify better the morphological differences between KO and control groups regarding myod expression in myotomes. Specify the number of animals analyzed.

Comments on the Quality of English Language

Some light revisions are needed to enhance the clarity and correctness of the English language.

Reviewer 2 Report

Comments and Suggestions for Authors

In this study Zhou et al studied the impact of lncRNA gas5 on the development of zebrafish embryos. This analysis was based on a sequence analysis of the transcriptome of developmental stages 10 hpf, 24 hpf and 36 hpf.

The manuscript is mainly about one lncRNA (gas5). I don’t think the title really fits, since gas5 was found by cherry picking and not by systematic identification. Also, the part about gene functional analysis does not fit to the title.

In general, the manuscript is very unfocused and apart from gas5 I can see no guiding thread.

The topic of this manuscript is of potential interest but of limited originality. In the current version I don’t think that the manuscript can be recommended for publication in IJMS.

Major concerns

1.     Which aligner was used for the read alignment? How did you calculate gene / transcript expression?

2.     Figure S1: Please give the read depth for all samples. A table would be sufficient. This is relevant to assess reliability of transcripts with a low read count. Did the authors set a read count threshold for ‘unexpressed’ genes? If not, I think that the number of reliably regulated transcripts is highly overestimated. E.g. in table S1 more than 1000 genes show an expression < 1 and high logFCs in all four samples compared. For such low numbers a high absolute logFC is not meaningful.

3.     Statistical analysis, line 406: ‘The qPCR experiment was carried out in triplicate’ If n<9 data can not be considered normally distributed. Therefore t-test is not appropriate. Please use a non-parametrical test. It is not mentioned in the methods part, which test has been used for the data displayed in Figure 3 and S3 / S4. But, since this is a comparison with more than 2 groups even though all groups are compared against time point 0 or 10, please use a suitable test for the comparison of three or more unmatched groups (non-parametric) with appropriate post-hoc test and describe this in the methods.

4.     Line 223ff:’ …, after lncRNA gas5 was knocked out using 223 CRISPR/Cas9 technology, qPCR was used to detect the expression of its five neighboring 224 genes (tor3a, osbpl9, calr, si:ch211-198n5+11, rpe65b) in zebrafish 24 hpf embryos (Figure 5b, 225 Supplementary table 16).‘  In table S16, there are ‘The target genes of lncRNAs’ listed, not  neighboring genes. Further I can find no explanation in the methods, how these targets were identified. What does ‘distance’ in the table headers mean? Distance from gas5 on the chromosome? Are all genes on the same chromosome as the lncRNA targeting them?

5.     Parts describing the methods should not be in the results part (e.g. lines 84-91; lines 130-136; lines 165-169; …), but in the methods part.

6.     Figure 3a:  time point 96 hpf is missing.

7.     Figure S6 / lines 172ff.: To me the CRISPR/Cas9 line looks rather like a mixture of KO and wild-type sequences. It might be rather comparable to a knock down. Can this be the reason for differing results in figure 6? How many individuals have been submitted to sequencing? Please state in the methods.

8.     Please describe in the methods, which RNA you used for qPCR. Is it the same that was used for sequencing or did you prepare new RNA (n=?).

9.     Figure S3 / S4 says in the y-axis ‘relative expression’. For RNA-seq, relative to what?

10.  In this study, three timepoints are analyzed. The comparisons of two groups are helpful, but not sufficient. In addition, please use some tools for the analysis of time courses (e.g. R/Bioconductor package ‘mfuzz’).

11.  Lines 264-265: ‘And genome-wide sequencing was performed to obtain genes associated with specific organogenesis processes, …’ A general part about specific organogenesis processes is missing in the manuscript. Further, the manuscript states to be about lncRNAs and not genes.

12.  To a great extend the discussion is redundant to the introduction and the result part. This should be rewritten and the results / focus on muscle development … better discussed.

Comments on the Quality of English Language

Minor editing of English language required

Round 2

Reviewer 1 Report

Comments and Suggestions for Authors

The authors responded to most of the comments I highlighted and the paper is now suitable for publication

Reviewer 2 Report

Comments and Suggestions for Authors

Most of the questions I was asking were not adequately answered. E.g. In remark 1 I was asking about the aligner and not to which genome the reads were aligned to and how (which program) the authors used to calculate expression, remark 9 was not about qPCR but RNA seq etc... implicating the authors do not understand the questions. Further, I still think the statistics is not done in a reliable and meaningful way and the analysis should be improved (as proposed in remark 10, but not answered in any context to my remark).

Therefore I think the manuscript has not been substantially improved.

Round 3

Reviewer 2 Report

Comments and Suggestions for Authors

I think that most point have been clarified, yet there are some concerns:

1.      The authors did not answer my question regarding figure S6:

‘7. Figure S6 / lines 172ff.: To me the CRISPR/Cas9 line looks rather like a mixture of KO and wild[1]type sequences. It might be rather comparable to a knock down. Can this be the reason for differing results in figure 6? How many individuals have been submitted to sequencing? Please state in the methods. Response: Figrue S6 is the result of 24hpf embryo sequencing.’

I think this is relevant to evaluate the ko.

2.      Lines 336-342: The authors describe the QC of the reads (lines 335-341). ‘… After qualification, sequencing was performed using DNBSEQ- 341 T7 sequencer (MGI Tech Co., Ltd. China) with PE150 model.’ This sentence is either misplaced (place before the new paragraph) or wrong (what means ‘after qualification’?)

3.      STAR software not ‘STRA software‘

4.      Line 347: please use ‚differentially expressed genes’ instead of ‘differentially genes’

5.      Line 347: Cis targets were predicted that genes within 100kb upstream and downstream of the lncRNA locus were selected as candidate target genes.‘ I do not understand the sentence. Please correct.

6.      I don’t think, that the additional files 1-6 are helpful for the presentation of 4 samples, since values cannot be compared between genes and due to the z-score normalization numbers are not informative.

Author Response

Comments and Suggestions for Authors

I think that most point have been clarified, yet there are some concerns:

  1. The authors did not answer my question regarding figure S6:

‘7. Figure S6 / lines 172ff.: To me the CRISPR/Cas9 line looks rather like a mixture of KO and wild[1]type sequences. It might be rather comparable to a knock down. Can this be the reason for differing results in figure 6? How many individuals have been submitted to sequencing? Please state in the methods. Response: Figrue S6 is the result of 24hpf embryo sequencing.’

I think this is relevant to evaluate the ko.

Response: Figure 5 showed the Sanger sequencing map of 8 24hpf embryos. ‘NI’ represents the sequencing map of 8 24hpf embryos with nothing injection, ‘control’ represents that of 8 embryos with only injection of Cas9 protein, ‘CRISPR/Cas9’ represents that of 8 embryos with coinjection of sgRNA and Cas9 protein. The sequencing map of 8 embryos with coninjection of sgRNA and Cas9 protein showed cross-peaks, but the other don’t, which indicated there were sequence mutant embryos.

   In conclusion, this is relevant to evaluate the KO.

  1. Lines 336-342: The authors describe the QC of the reads (lines 335-341). ‘… After qualification, sequencing was performed using DNBSEQ- 341 T7 sequencer (MGI Tech Co., Ltd. China) with PE150 model.’ This sentence is either misplaced (place before the new paragraph) or wrong (what means ‘after qualification’?)

Response: Have been corrected

  1. STAR software not ‘STRA software‘

Response: Have been corrected

  1. Line 347: please use ‚differentially expressed genes’ instead of ‘differentially genes’

Response: Have been corrected.

  1. Line 347: ‘Cis targets were predicted that genes within 100kb upstream and downstream of the lncRNA locus were selected as candidate target genes.‘ I do not understand the sentence. Please correct.

Response: Have been corrected.

  1. I don’t think, that the additional files 1-6 are helpful for the presentation of 4 samples, since values cannot be compared between genes and due to the z-score normalization numbers are not informative.

Response: The additional files 1-6 have been removed.